# Protein Arginine Methylation Patterns in Plasma Small Extracellular Vesicles Are Altered in Patients with Early-Stage Pancreatic Ductal Adenocarcinoma

**DOI:** 10.3390/cancers16030654

**Published:** 2024-02-03

**Authors:** Kritisha Bhandari, Jeng Shi Kong, Katherine Morris, Chao Xu, Wei-Qun Ding

**Affiliations:** 1Department of Pathology, University of Oklahoma Health Sciences Center, Oklahoma City, OK 73104, USA; kritisha-bhandari@ouhsc.edu (K.B.); jengshi-kong@ouhsc.edu (J.S.K.); 2Department of Surgery, University of Oklahoma Health Sciences Center, Oklahoma City, OK 73104, USA; 3Department of Biostatistics & Epidemiology, University of Oklahoma Health Sciences Center, Oklahoma City, OK 73104, USA; chao-xu@ouhsc.edu

**Keywords:** small extracellular vesicles, pancreatic ductal adenocarcinoma, arginine methylation, biomarkers, symmetric dimethylarginine

## Abstract

**Simple Summary:**

Sensitive and specific circulating biomarkers for the early detection of pancreatic ductal adenocarcinoma (PDAC) are urgently needed to improve the survival outcomes of this malignancy. Small extracellular vesicles (sEVs) from cancer cells carry biomolecules of cellular origin that can be released into the circulation. Studies have shown that plasma sEV molecules, such as proteins and microRNAs, are potential indicators of PDAC. However, post-translational modifications of plasma sEV proteins, such as arginine methylation patterns, have never been examined as potential circulating biomarkers for PDAC. Protein arginine methylation is considered a relatively stable post-translational modification, and is a newly established molecular feature of PDAC. We thus speculated that arginine methylation patterns in plasma sEVs are non-invasive biomarkers for pancreatic ductal adenocarcinoma. In this report, we demonstrate that protein arginine methylation patterns are altered in plasma sEVs derived from patients with early-stage PDAC, with these findings supporting the development of these patterns as biomarkers for PDAC.

**Abstract:**

Small extracellular vesicles (sEVs) contain lipids, proteins and nucleic acids, which often resemble their cells of origin. Therefore, plasma sEVs are considered valuable resources for cancer biomarker development. However, previous efforts have been largely focused on the level of proteins and miRNAs in plasma sEVs, and the post-translational modifications of sEV proteins, such as arginine methylation, have not been explored. Protein arginine methylation, a relatively stable post-translational modification, is a newly described molecular feature of PDAC. The present study examined arginine methylation patterns in plasma sEVs derived from patients with early-stage PDAC (*n* = 23) and matched controls. By utilizing the arginine methylation-specific antibodies for western blotting, we found that protein arginine methylation patterns in plasma sEVs are altered in patients with early-stage PDAC. Specifically, we observed a reduction in the level of symmetric dimethyl arginine (SDMA) in plasma sEV proteins derived from patients with early- and late-stage PDAC. Importantly, immunoprecipitation followed by proteomics analysis identified a number of arginine-methylated proteins exclusively present in plasma sEVs derived from patients with early-stage PDAC. These results indicate that arginine methylation patterns in plasma sEVs are potential indicators of PDAC, a new concept meriting further investigation.

## 1. Introduction

Pancreatic ductal adenocarcinoma (PDAC) is a lethal neoplasm with a poor 5-year overall survival rate of 11% [1]. According to the data from the National Cancer Institute’s Surveillance, Epidemiology, and End Results (SEER), approximately 52% of PDAC patients are diagnosed when the cancer has already metastasized. The overall survival is much improved for patients with localized tumors, indicating that early detection of PDAC is key to improve its prognosis. Despite many attempts to identify biomarkers for PDAC [2,3,4], there are no reliable circulating biomarkers currently available for early detection of the disease. New strategies for PDAC biomarker development are desperately needed in order to enable early detection and improve PDAC patient survival.

Recent advancements in cancer biology revealed that small extracellular vesicles (sEVs) contain biomolecules of cellular origin, including lipids, proteins and nucleotides [5,6]. The transfer of sEVs from primary tumors to the circulation has been documented in various model systems [7,8]. We and others have demonstrated that the levels of plasma sEV molecules, including proteins and microRNAs, are potentially indicative of early-stage PDAC [9,10,11,12,13,14,15]. However, current efforts to investigate plasma sEV-based biomarkers are focused on the level of proteins or microRNAs present in the sEVs [16,17], and the potential of plasma sEVs for biomarker development remains to be further explored. Notably, post-translational modifications of plasma sEV proteins, such as arginine methylation patterns, have not previously been studied as circulating biomarkers for PDAC.

Protein arginine methylation refers to the transfer of a methyl group to the arginine residues of proteins, which is catalyzed by protein arginine methyltransferases (PRMTs). Type I PRMT enzymes catalyze the addition of two methyl groups to the same guanidine nitrogen of arginine, yielding asymmetric dimethylarginine (ADMA). Type II PRMTs mediate the addition of the second methyl group to another nitrogen of arginine, generating symmetric dimethylarginine (SDMA). All PRMTs catalyze the transfer of one methyl group to one of the guanidine nitrogens of arginine, producing mono-methylarginine (MMA) [18,19]. Altered protein arginine methylation is involved in PDAC pathogenesis and progression [19,20,21], and the expression level of most of the PRMTs is significantly associated with PDAC patient survival rates [22], indicating a new molecular feature of PDAC. Considering that cancer sEVs are released into the circulation and are detectable using various assays [14,23], the protein arginine methylation patterns of PDAC are likely detectable in the circulation, evoking the idea that arginine methylation patterns in plasma sEVs possess the potential to be novel circulating biomarkers for this malignancy.

In the present study, we characterized arginine methylation patterns in plasma sEVs using western blot and proteomics analysis. We found that arginine methylations of plasma sEV proteins are altered in patients with early-stage PDAC compared with that in healthy controls or patients with chronic pancreatitis, strongly supporting the development of arginine methylation patterns in plasma sEVs as indicators of early-stage PDAC.

## 2. Materials and Methods

### 2.1. Cell Lines

Pancreatic cancer cell lines PANC-1 and MIA PaCa-2, and the pancreatic ductal cell line hTERT-HPNE, were purchased from ATCC and cultured in the recommended cell culture medium, supplemented with exosome-depleted FBS in a humidified environment at 37 °C and 5% CO_2_.

### 2.2. Patient Plasma

Patient plasma samples were collected under an approved IRB protocol (IRB#: 5535). Healthy plasma was provided by the Oklahoma Blood Institute. Plasma from colon cancer patients was collected at the Stephenson Cancer Center. All other plasma samples were obtained from the NCI-sponsored Cooperative Human Tissue Network (CHTN), which collects patient plasma samples following standard protocols. The experiments were performed using age- and gender-matched plasma samples among the groups, the majority of which were also matched for their race (Appendix A).

### 2.3. Cellular and Plasma sEV Isolation

The isolation of cell line- and human plasma-derived sEVs was performed using double filtration followed by total sEV isolation using a commercial kit, as we previously described [24]. Cells were cultured in 20 mL exosome-depleted media and the media was harvested upon 80% cell confluency. The media was centrifuged at 10,000× *g*, 4 °C to remove cellular debris, followed by ultrafiltration using 100 kDa filters (Amico Ultra-15, Merk Millipore Ltd., Burlington, MA, USA). The media was further subjected to filtration using 0.22 µm and 0.1 µm PVDF filters (Ultrafree-CL, Merk Millipore Ltd., UFC40GV00 and UFC40VV00). The filtrate obtained was precipitated using the total exosome isolation kit (Invitrogen, Carlsbad, CA, USA), overnight at 4 °C. The precipitate was centrifuged at 10,000× *g* for 1 h at 4 °C and the pellet was lysed using RIPA buffer (Sigma-Aldrich, Saint Louis, MO, USA) containing 1× protease inhibitor cocktail (Pierce protease inhibitor A32953 and phosphatase inhibitor A32957). BCA was performed to analyze the protein concentration for downstream analysis. For plasma sEV isolation, 100 µL of plasma was precleared by centrifugation at 3000× *g* for 5 min at room temperature followed by fibrin precipitation using 5 U/mL thrombin. The fibrin pellet was removed upon centrifugation at 10,000× *g* for 5 min and the supernatant was diluted with 1× PBS in a 1:4 ratio and subjected to filtration by 0.22 µm and 0.1 µm PVDF filters consecutively. The filtrate was subjected to sEV precipitation using the total exosome isolation kit (Invitrogen, 4478359) as described above.

### 2.4. Western Blot Analysis

A total of 50 ug of proteins per sample were loaded onto 10% SDS gel, and transferred to the PVDF membrane at 100 V for 2 h. The membranes were blocked in 5% non-fat dry milk in 1× TBST for 1 h at 4 °C and probed with primary antibodies at the dilution of 1:1000 for GAPDH (Cell signaling technology, Danvers, MA, USA, 2118S), SDMA (Cell signaling technology, 13222S), MMA (Cell signaling, 8015S), ADMA (Cell signaling, 13522), Complement C3 (Santa Cruz Biotechnology, Dallas, TX, USA, sc-28294) and Alpha-2-macroglobulin (Santa Cruz Biotechnology, sc-390544) overnight, at 4 °C. The membranes were imaged for 10 min (for sEV lysates) or 2 min (for cellular lysates) using the LiCOR Odyssey Fc imaging system (Linclon, NE, USA). The protein bands detected were quantified using Adobe Photoshop Elements 6.0 for the determination of relative band intensity.

### 2.5. Proteomics

Proteomic analysis was performed at the Proteomics core, University of Oklahoma Health Sciences Center. For the identification of the proteins detected in bands I and II of SDMA, the plasma sEV proteins were lysed using RIPA buffer containing protease inhibitor cocktail. The lysates were separated on a 10% SDS gel and Coomassie staining of the gel was performed. The gel was excised at the sizes corresponding to the SDMA bands I and II, and analyzed at the Proteomics core using a Thermo Fusion Lumos Tribrid orbitrap coupled to an ultra-performance nanoscale capillary liquid chromatography (LC/ESI/MS/MS) to achieve unbiased qualitative and quantitative analyses. The proteomics data were further analyzed using the tandem mass spectrometry data analysis program Sequest.

For proteomic profiling of the arginine methylated peptides in plasma sEVs, the isolated plasma sEV lysates (200 µg) were added to the IP buffer (q.s. 200 uL), precleared for 1 h at 4 °C with Dynabeads, incubated with the SDMA antibody (1:50 dilution, Cell signaling, 13222S) overnight at 4 °C and loaded onto the Dynabeads Protein G (Invitrogen, 10004D) [25] and incubated for 20 min at room temperature. The beads were washed 5 times with IP buffer before elution of the proteins with 50 mM glycine, pH 2.8. The eluted proteins were digested with trypsin, reduced with 10 mM dithiothreitol and alkylated with 10 mM iodoacetamide. The peptides were dried, resuspended and analyzed using a Thermo Fusion Lumos Tribrid orbitrap coupled to an ultra-performance nanoscale capillary liquid chromatography (LC/ESI/MS/MS). Raw MS data were processed by PLGS (ProteinLynx Global Server, Waters Corp., Manchester, UK) for peptide and protein identification. The MS/MS spectra were searched against the Uniprot Human database with the carbamidomethylation of cysteine residues and methylation of arginine residues set as fixed modifications.

### 2.6. Immunoprecipitation

A total of 200 ug of precleared protein lysates were immunoprecipitated using Dynabeads Protein G (Invitrogen, 10004D) following the manufacturers’ instructions. Antibodies against SDMA, MMA and normal rabbit IgG (1:50 dilution) were used to immunoprecipitate plasma sEV proteins. The immunoprecipitated proteins were loaded directly onto the 10% SDS gel and analyzed by western blot to detect Complement C3 and Alpha-2-macroglobulin, or the immunoprecipitated proteins were submitted for proteomics profiling as described above.

### 2.7. Bioinformatics

Arginine methylation site prediction for Complement C3 and Alpha-2-macroglobulin was performed using the online tool PRmePRed (http://bioinfo.icgeb.res.in/PRmePRed/, accessed on 12 May 2022). DAVID, a web-based bioinformatics tool [26], was utilized to cluster the data obtained from proteomics.

### 2.8. Statistical Analysis

GraphPad Prism 10 was used to analyze the data obtained from the Western blot quantification. One-way ANOVA, followed by Dunnett’s multiple comparisons, was applied to determine the differences among experimental groups. *p* < 0.05 was considered a significant difference.

## 3. Results

### 3.1. Detection of Protein Arginine Methylation in Cellular and sEV Lysates

For the initial part of the study, antibody validation was performed on cellular and sEV lysates obtained from pancreatic cancer cell lines (PANC-1 and MIA PaCa-2) and the pancreatic ductal cell line hTERT-HPNE (Figure 1A–D) using anti-SDMA and anti-MMA antibodies. Multiple bands were detected in all of the lysates, indicating the diversity of substrates of PRMTs in these cells. The degree of arginine methylation (both MMA and SDMA) was lower in the lysates derived from the pancreatic ductal cell line HPNE. These results indicate the possibility of detecting variations in the plasma sEV protein arginine methylation between patients with PDAC and healthy controls.

### 3.2. SDMA Levels in Plasma sEVs Are Reduced in Patients with Early- and Late-Stage PDAC but Remain Unchanged in Patients with Chronic Pancreatitis

We isolated plasma sEVs using double filtration followed by precipitation using a total exosome isolation kit as previously described [24]. The isolated sEVs were examined by western blotting to detect the exosome surface markers Flotillin-1, CD63 and CD9 and the negative marker Calnexin (Figure 1E). These results showed that the isolated sEVs were predominately exosomes. The particle numbers and sizes of the isolated sEVs were analyzed using nanoparticle tracking analysis (Figure 1F, Nanosight NS300 System, Malvern Instruments, Malvern, UK).

Five groups of plasma were used in this study: plasma from patients with stage I-II PDAC (Table 1), stage III–IV PDAC, stage I-III colon cancer, and chronic pancreatitis (Appendix A), and gender- and age-matched healthy subjects (Table 2). Western blot analysis was performed to assess the arginine methylation patterns of plasma sEVs using the arginine methylation-specific antibodies MMA, SDMA, and ADMA. Three bands (band I, II and III) were consistently detected across all samples for the three arginine methylation-specific antibodies. (Figure 2A–C). Coomassie blue staining of the gel was performed to serve as protein loading control.

Semi-quantification of the Western blot bands demonstrated that the intensities of the top two bands in SDMA detection were reduced up to 40% in patients with early-stage PDAC and colon cancer compared with that of matched healthy subjects (Figure 2E). However, the intensities of the corresponding MMA and ADMA bands were only altered in patients with colon cancer but not patients with early-stage PDAC (Figure 2D,F).

Similar experiments were performed using plasma samples from patients with early-stage PDAC, late-stage PDAC and chronic pancreatitis. A significant reduction in SDMA levels (band I and band II) in plasma sEVs was detected in patients with early-stage PDAC and late stage PDAC compared with that of healthy subjects (Figure 3), whereas plasma sEV SDMA levels remain unchanged in patients with chronic pancreatitis (Figure 4), indicating different arginine methylation patterns in plasma sEVs among the groups of patients. No significant reduction in plasma sEV MMA levels was detected in plasma sEVs from all patient groups except those from late-stage PDAC (Figure 3 and Figure 4).

### 3.3. Complement C3 and Alpha-2-Macroglobulin Are Major Proteins Identified in the Top Two Bands of SDMA Detection

Since the intensities of the top two SDMA bands detected by western blot were significantly reduced in plasma sEVs derived from patients with early-stage PDAC, we opted to identify the proteins detected in these two SDMA bands. The two bands on the G-250-stained gel were cut at around the molecular weight of 180 kDa for band I and 100–135 kDa for band II and submitted to proteomics analysis (Appendix A). A Thermo Fusion Lumos Tribrid orbitrap coupled to an ultra-performance nanoscale capillary liquid chromatograph (LC/ESI/MS/MS) was used to achieve unbiased qualitative and quantitative analyses. The proteomics data were further analyzed using the tandem mass spectrometry data analysis program Sequest (Figure 5A,B). Many proteins were detected for the two bands (Figure 5C). However, bioinformatics analysis using DAVID [26] revealed that 70.88% of band I and 66.15% of band II were associated with sEVs (exosomes, Appendix A).

Out of the proteins identified from bands I and II, Complement C3 and Alpha-2-macroglobulin were recognized with the highest abundance and Sequest score (Figure 5A,B). These two proteins were abundantly present in the plasma and have been previously detected in sEVs [27,28,29]. Interestingly, the expression of Complement C3 and Alpha-2-macroglobulin was reported to be upregulated in PDAC tissues and patient plasma [30,31,32]; however, arginine methylation of the two proteins has never been previously described in any tissue specimens and cell lines.

### 3.4. Complement C3 and Alpha-2-Macroglobulin Harbor Arginine Methylation Sites

We utilized the online tool PRmePRed (http://bioinfo.icgeb.res.in/PRmePRed/, accessed on 30 January 2024) to predict arginine methylation sites in Complement C3 and Alpha-2-macroglobulin [33]. PRmePRed identified six possible arginine methylation sites with a prediction score of >0.8 (Figure 6A,B) for both of the proteins, indicating the presence of arginine methylation sites in Complement C3 and Alpha-2-macroglobulin. To experimentally confirm this prediction, we immunoprecipitated plasma sEV proteins with antibodies against SDMA or MMA and detected Complement C3 and Alpha-2-macroglobulin in the immunoprecipitated fraction using western blot analysis, with the level of Complement C3 and Alpha-2-macroglobulin being more pronounced in the SDMA precipitants (Figure 6C,D). The antibodies for Complement C3 and Alpha-2-macroglobulin detection were validated using plasma sEVs (Appendix A). These observations support the presence of methylated arginine sites in these two proteins.

### 3.5. An Unbiased Proteomic Analysis of Plasma sEV Arginine-Methylated Proteins Isolated from Patients with Early-Stage PDAC and Matched Healthy Subjects

To further determine whether the arginine methylation patterns in plasma sEVs differ between patients with early-stage PDAC and matched healthy subjects, we performed an unbiased proteomic analysis of the isolated plasma sEVs. Arginine methylated proteins in plasma sEVs were enriched by immunoprecipitation using an SDMA antibody conjugated to magnetic beads. The immunoprecipitation with the SDMA antibody was validated using PANC-1 cell lysates (Figure 7A) and plasma-derived sEV proteins (Figure 7B) by western blot analysis, detecting multiple bands in the immunoprecipitated fractions. The SDMA antibody pull-down fractions of plasma sEVs were subjected to LC/ESI/MS/MS profiling, and arginine-methylated proteins were identified. There were 61 arginine-methylated proteins detected in healthy plasma sEVs and 69 detected in PDAC plasma sEVs. Interestingly, only 16 arginine-methylated proteins were shared by the two groups, and the rest were exclusively present in either healthy or PDAC plasma sEVs, indicating the strong potential of these arginine-methylated proteins as biomarkers for the early detection of PDAC. Some of the proteins exclusively detected in PDAC plasma sEVs are closely associated with cancer progression (Figure 7C), thus representing the most interesting candidate proteins, based on arginine methylation, for PDAC biomarker development.

## 4. Discussion

PDAC is the most lethal form of pancreatic cancer and has been predicted to become the second leading cause of cancer related death soon [34]. Early detection is key to improving PDAC patient survival [1]. Plasma CA19-9 (carbohydrate antigen) has been frequently used in the clinic to detect PDAC, but its sensitivity and specificity remains unsatisfactory [35]. There is an urgent need for the development of robust circulating biomarkers for PDAC. The results from the present study demonstrate that arginine methylation patterns in plasma sEVs are altered in patients with early-stage PDAC, indicating for the first time that some of the arginine-methylated proteins in the plasma sEVs are potential biomarkers for the early detection of PDAC. Furthermore, while plasma sEVs have been explored for biomarker development for human cancers [10,14,15,23,36], arginine methylation patterns in plasma sEVs have never been previously characterized in any cancer types. Thus, our findings may have general implications and open up a new avenue in biomarker development for various malignancies.

Protein methylation is one of the most abundant post-translational modifications observed experimentally [37]. It is considered a relatively stable post-translational modification [18,38], and may serve as a tag of the protein, favoring its application for biomarker development. Characterization of arginine methylation using proteomics was a challenge in the past [39,40]. However, with the development of antibodies specific to protein arginine methylation, the enrichment of arginine-methylated peptides and detection of protein arginine methylation have become possible [39]. Biochemical methods for the detection of arginine methylation are well established [41,42,43,44,45], including antibody-based and chemical-based approaches to analyze arginine-methylated peptides in cells [43], tissues [46] and human plasma [45]. In the present study, we utilized antibodies against SDMA, MMA or ADMA to detect arginine methylation in plasma sEVs by western blot or to enrich arginine-methylated peptides from plasma sEVs by immunoprecipitation for proteomic analysis. Two key findings are presented: first, we demonstrate that the SDMA level, specifically bands I and II, was significantly lower in plasma sEVs derived from patients with early-stage PDAC compared with the SDMA level in plasma sEVs from matched healthy subjects and patients with chronic pancreatitis. Although the plasma sEV SDMA level was also lower in patients with colon cancer, the difference between patients with colon cancer and those with PDAC was noted in their plasma sEV ADMA and MMA levels (Figure 2D,F), suggesting a unique arginine methylation pattern in PDAC plasma sEVs. Second, using an unbiased proteomic approach, we observed that a group of arginine-methylated proteins are exclusively present in plasma sEVs derived from patients with early-stage PDAC compared with those from matched healthy subjects. The exclusive presence of some of the arginine methylated proteins in PDAC plasma sEVs strongly supports the notion that arginine methylation patterns in plasma sEVs are indicators of early-stage PDAC. Among the 54 proteins exclusively present in plasma sEVs from patients with early-stage PDAC, 8 proteins were noted for their close association with human cancer (Figure 7C), and this list of proteins will be prioritized in our future efforts in developing circulating biomarkers for the early detection of PDAC. Furthermore, by expanding our sample size, a model of arginine methylation profiling in plasma sEVs could be established to predict the presence of early-stage PDAC.

Another interesting observation from this study is that we identified Complement C3 and Alpha-2-macroglobulin to represent the major proteins detected in bands I and II of SDMA in plasma sEVs. Both of the proteins are abundant in plasma and have been detected in sEVs derived from human cells [27,28,29]. Their levels in the circulation have been reported to be potential diagnostic markers for PDAC [31,32,47]. However, the arginine methylation status of these proteins has never been described. Using the PRmePRed online tool [33], we were able to identify highly probable arginine methylation sites in Complement C3 and Alpha-2-macroglobulin proteins. This was further confirmed by the immunoprecipitation of plasma sEV lysates with an SDMA or MMA antibody and detection of Complement C3 and Alpha-2-macroglobulin in the immunoprecipitated fractions. Our results indicate that not only are the levels of these two proteins in the circulation potential biomarkers for PDAC as previously reported [31,32,47], their arginine methylation status in plasma sEVs is also likely indicative of early-stage PDAC, which merits further investigation.

The limitations of the present study include a relatively small sample size and its retrospective nature. We envision that large cohort studies, both retrospective and prospective, are needed to verify our findings and establish arginine methylation patterns in plasma sEVs as biomarkers for early detection of PDAC.

## 5. Conclusions

In summary, the present study has demonstrated the alterations of arginine methylation patterns in plasma sEVs derived from patients with early-stage PDAC and identified arginine-methylated proteins that are exclusively present in PDAC plasma sEVs. These findings support the view that arginine methylation patterns in plasma sEVs are potential circulating biomarkers for the early detection of PDAC, a concept that has not been previously examined.

## Figures and Tables

**Figure 1 cancers-16-00654-f001:**
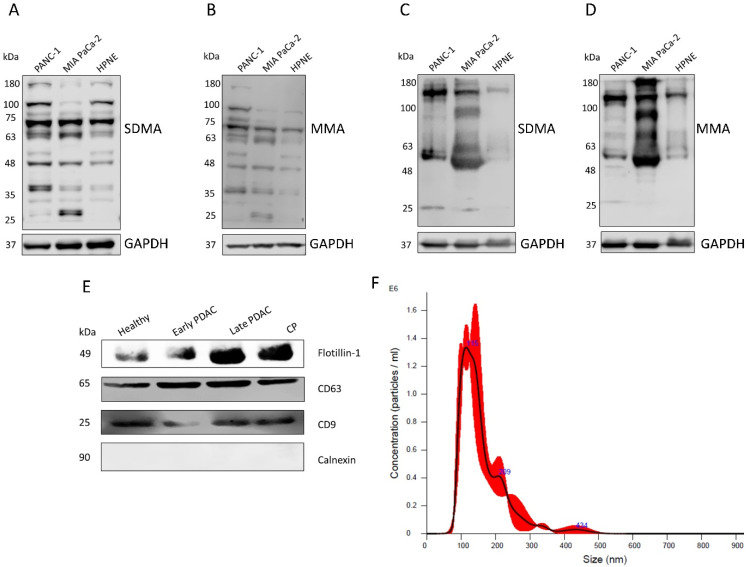
Validation of the arginine methylation antibodies and the sEVs isolated from human plasma. (**A**,**B**) Western blot detection of SDMA and MMA in PANC-1, MIA PaCa-2 and HPNE cell lysates. (**C**,**D**) Western blot detection of SDMA and MMA in PANC-1, MIA Paca-2 and HPNE cell derived sEV lysates. (**E**) Western blot detection of exosome markers (Flotillin-1, CD63 and CD9) and the negative marker, Calnexin, in human plasma sEVs. (**F**) Particle size determination of plasma sEVs using nanoparticle tracking analysis (NTA).

**Figure 2 cancers-16-00654-f002:**
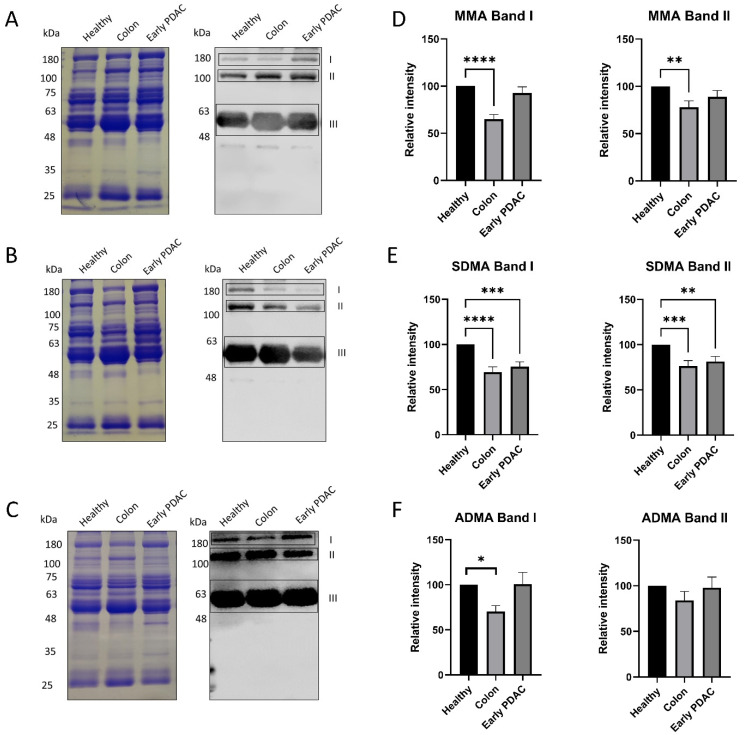
Arginine methylation patterns in plasma sEVs derived from healthy subjects and patients with PDAC and colon cancer. Western blot analysis of individual plasma sEV lysates was performed using antibodies against MMA, SDMA and ADMA. Representative gels are shown. (**A**) MMA detection: (**left**), Coomassie blue staining of the gel as loading control; (**right**), detection of MMA. (**B**) SDMA detection: (**left**), Coomassie blue staining of the gel as loading control; (**right**), detection of SDMA. (**C**) ADMA detection: (**left**), Coomassie blue staining of the gel as loading control; (**right**), detection of ADMA. (**D**) Quantification of the MMA band I and band II. (**E**) Quantification of SDMA band I and band II. (**F**) Quantification of the ADMA band I and band II. **** *p* < 0.0001, *** *p* < 0.001, ** *p* < 0.01, * *p* < 0.05, one-way ANOVA followed by Dunnett’s multiple comparisons (*n* = 16).

**Figure 3 cancers-16-00654-f003:**
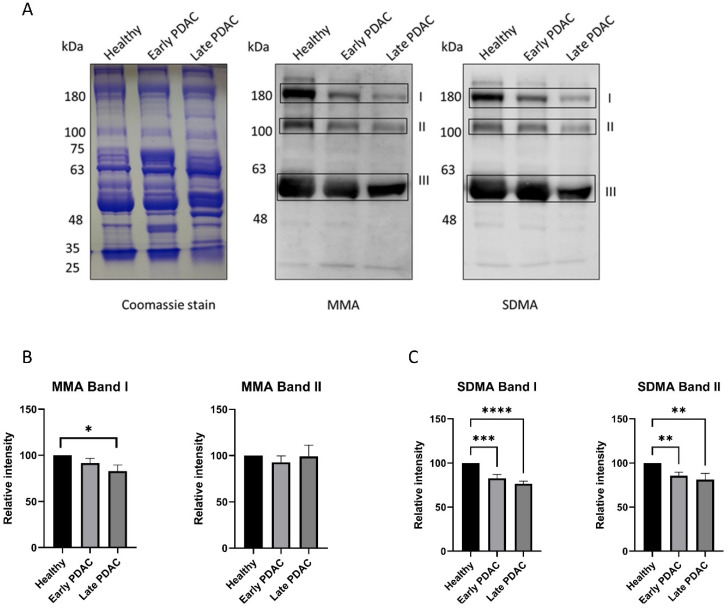
Arginine methylation patterns in plasma sEVs derived from healthy subjects and patients with early- and late-stage PDAC. Western blot of individual plasma sEV lysates was performed using antibodies against MMA and SDMA. Representative gels are shown. (**A**) MMA and SDMA detection in plasma sEVs derived from patients with early- and late-stage PDAC: (**left**), Coomassie blue staining of the gel as loading control; (**middle**), detection of MMA; (**right**), detection of SDMA. (**B**) Quantification of MMA bands I and II. (**C**) Quantification of SDMA bands I and II. **** *p* < 0.0001, *** *p* < 0.001, ** *p* < 0.01, * *p* < 0.05, one-way ANOVA followed by Dunnett’s multiple comparisons (*n* = 23 for early-stage PDAC and 22 for healthy subjects, *n* = 10 for late-stage PDAC).

**Figure 4 cancers-16-00654-f004:**
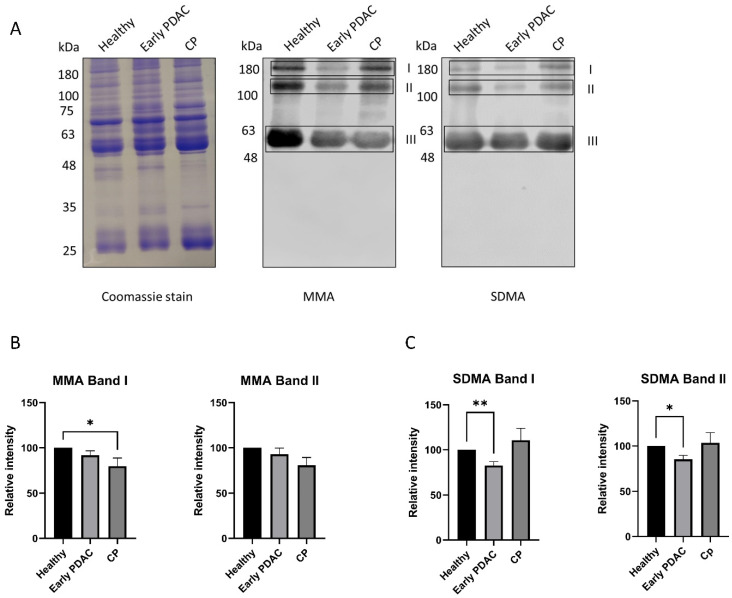
Arginine methylation patterns in plasma sEVs derived from healthy subjects and patients with early-stage PDAC and chronic pancreatitis (CP). Western blot of plasma sEV lysates was performed using antibodies against MMA and SDMA. Representative gels are shown. (**A**) MMA and SDMA detection: **left**, Coomassie blue staining of the gel as loading control; **middle**, detection of MMA; **right**, detection of SDMA. (**B**) Quantification of MMA band I and II. (**C**) Quantification of SDMA band I and II. * *p* < 0.05, ** *p* < 0.01, one-way ANOVA followed by Dunnett’s multiple comparisons (*n* = 23 for early-stage PDAC and 22 for healthy subjects, and *n* = 8 for CP).

**Figure 5 cancers-16-00654-f005:**
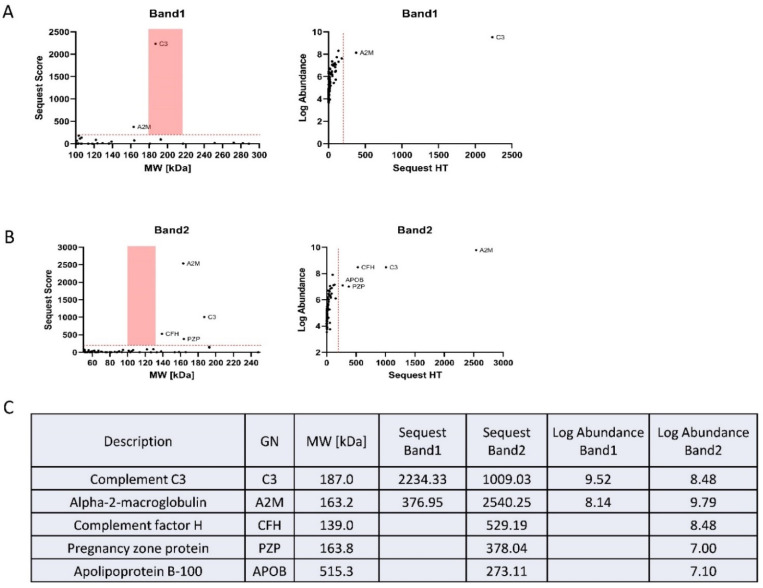
Identification of plasma sEV proteins corresponding to SDMA band I and band II. (**A**) The proteomics data obtained from digestion of gel portions corresponding to SDMA band I, and analyzed using Sequest: (**left**), Sequest score of the detected proteins; (**right**), abundance of the proteins. (**B**) The proteomics data obtained from the digestion of gel portions corresponding to SDMA band II, and analyzed using Sequest: (**left**), Sequest score of the detected proteins; (**right**), abundance of the proteins. (**C**) Proteins identified by Sequest analysis of the proteomics results.

**Figure 6 cancers-16-00654-f006:**
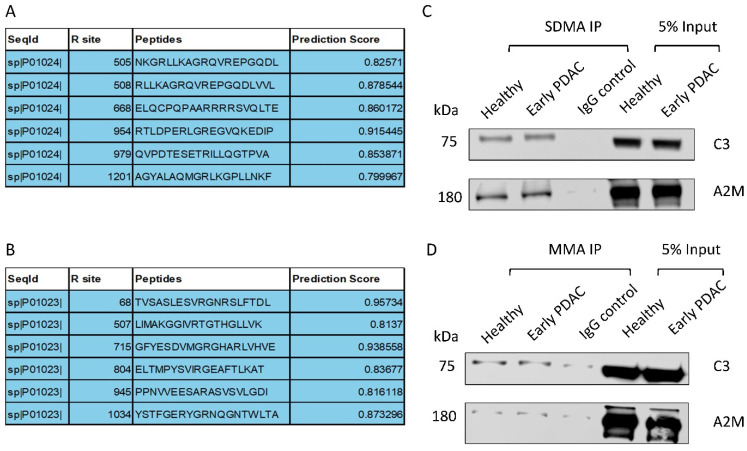
Validation of arginine methylation of Complement C3 and Alpha-2-macroglobulin. (**A**) Arginine methylation of Complement C3 and (**B**) Alpha-2-macroglobulin predicted by PRmePRed. (**C**,**D**) Western blot detection of Complement C3 and Alpha-2-macroglobulin in the immunoprecipitated fraction of plasma sEV lysates ((**C**), SDMA; (**D**) MMA).

**Figure 7 cancers-16-00654-f007:**
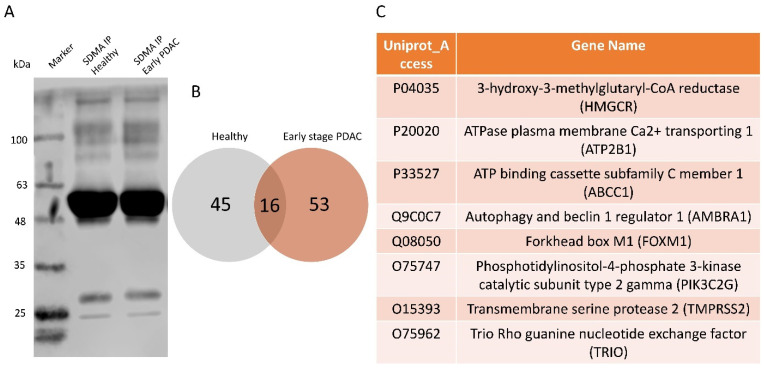
Proteomics analysis of the SDMA immunoprecipitated plasma sEV proteins from patients with early-stage PDAC and matched healthy subjects. Plasma sEV lysates (*n* = 18/group, pooled samples) from early-stage PDAC and healthy controls were immunoprecipitated with the SDMA antibody followed by proteomics analysis. (**A**) Western blot validation of immunoprecipitation by the SDMA antibody using PANC-1 cell lysates. (**B**) Venn-diagram showing numbers of arginine-methylated proteins that are shared or exclusively present in plasma sEV lysates from patients with early-stage PDAC (total 69 proteins) and healthy subjects (total 61 proteins). (**C**) Selected proteins that are exclusively arginine-methylated and closely associated with human cancer.

**Table 1 cancers-16-00654-t001:** Clinicopathological features of early-stage PDAC patients included in this study (*n* = 23, mean age: 60.69, median age: 60).

Plasma Sample	Age	Gender	Race	Tumor Stage ^a^
P1 ^¥,€^	74	Female	CA	1A
P2 ^¥,€^	43	Female	CA	1B
P3 ^¥,€^	50	Female	CA	1B
P4 ^¥,€^	39	Male	CA	2A
P5 ^¥,€^	55	Male	CA	1B
P6 ^¥,€^	59	Male	CA	2A
P7 ^¥,€^	36	Female	CA	2A
P8 ^¥,€^	76	Female	CA	2A
P9 ^¥,€^	67	Female	CA	2A
P10 ^¥,€^	55	Female	CA	2A
P11 ^¥,€^	80	Male	CA	1A
P12 ^¥,€^	69	Male	CA	1A
P13 ^¥,€^	52	Male	CA	1B
P14 ^¥,€^	50	Male	CA	2A
P15 ^¥,€^	69	Male	CA	2A
P16 ^¥,€^	60	Male	CA	1B
P17 ^€^	80	Female	AA	1A
P18 ^€^	60	Female	AA	1B
P19 ^€^	60	Female	AA	1B
P20 ^€^	79	Male	CA	2B
P21 ^€^	51	Male	CA	1B
P22 ^€^	63	Male	CA	2B
P23 ^€^	69	Male	CA	1A

^a^ Pathologic TNM tumor staging, (American Joint Committee on Cancer Care 8th Edition). CA, Caucasian, AA, African American. ^¥^ Subjects used relating to heathy controls and colon cancer. ^€^ Subjects used relating to late PDAC and chronic pancreatitis.

**Table 2 cancers-16-00654-t002:** Features of healthy subjects included in this study (*n* = 22, mean age: 59.22, median age: 60).

Plasma Sample	Age	Gender	Race
H1 ^¥,€^	73	Female	CA
H2 ^¥,€^	40	Female	CA
H3 ^¥,€^	50	Female	CA
H4 ^¥,€^	43	Male	CA
H5 ^¥,€^	55	Male	CA
H6 ^¥,€^	59	Male	CA
H7 ^¥,€^	60	Female	CA
H8 ^¥,€^	40	Female	CA
H9 ^¥,€^	73	Female	CA
H10 ^¥,€^	68	Female	CA
H11^¥,€^	48	Female	CA
H12 ^¥,€^	54	Female	CA
H13 ^¥,€^	80	Male	CA
H14 ^¥,€^	68	Male	CA
H15 ^¥,€^	52	Male	CA
H16 ^¥,€^	45	Male	CA
H17 ^€^	66	Male	CA
H18 ^€^	60	Male	CA
H19 ^€^	71	Female	AA
H20 ^€^	71	Male	CA
H21 ^€^	62	Male	CA
H22 ^€^	65	Male	CA

CA, Caucasian, AA, African American, NA, not available. ^¥^ Subjects used relating to early PDAC and colon cancer. ^€^ Subjects used relating to late PDAC and chronic pancreatitis.

## Data Availability

The original contributions presented in the study are included in the article/Appendix A, further inquiries can be directed to the corresponding authors.

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
