# Peer review of "Protein Arginine Methylation Patterns in Plasma Small Extracellular Vesicles Are Altered in Patients with Early-Stage Pancreatic Ductal Adenocarcinoma"

_cancers, 2024, doi:10.3390/cancers16030654_

Round 1

Reviewer 1 Report

Comments and Suggestions for Authors

cancers-2827280

Title: Protein arginine methylation patterns in plasma small extracellular vesicles are altered in patients with early stage pancreatic ductal adenocarcinoma

In the current study, Bhandari and colleagues evaluated protein methylation patterns in circulating small extracellular vesicles from patients with early stage pancreatic cancer (N= 23) and matched controls. Authors found that symmetric dimethyl arginine (SDMA) in plasma sEVs was lower in patients with early or late stage PDAC; immunoprecipitation followed by proteomic analyses identified candidate arginine methylated proteins that were exclusively identified in sEVs from early-stage PDAC cases.

 Overall the study is interesting. However, enthusiasm is thwarted due to inadequate description of experimental conditions as well as lack of validation. 

Patient plasmas. Authors state that “plasma from colon cancer patients was collected at the Stephenson Cancer Center”. Did the authors mean to say pancreatic cancer? While Tables 1 and 2 are fine, it would be easier to compare characteristics of cases and controls in a singular table that summarizes numbers as well as sex, age, and race distributions. Moreover, per text provide in results section, authors additionally included samples from patients with locally advanced colon cancer as well as patients with chronic pancreatitis. The source of these specimens needs to be detailed in the methods section.

Statistical analyses. Authors should detail which post hoc test was used for pairwise analyses following ANOVA.

Figures 2-3. It is unclear if individual patient plasmas were used or if samples were pooled. Moreover, it is very difficult to ascertain how many samples were evaluated in the respective groups. This information is pivotal to make meaningful conclusions. Additionally, there are no error bars for the ‘healthy’ group, which suggests an N= 1. It is unclear how statistical significance would be derived in this setting. Individual data points show be provided for these representative bar plots. Figure C, band II appears to be oversaturated.

Figure 5. Proteins identified in SDMA band I and II. C3, CFH, A2M, and APOB are abundant plasma proteins and associated with inflammatory responses. This would suggest that these are not derived from cancer cells.

Validation of findings. Based on Figure 7C, there are several candidate proteins that are methylated and selectively identified in sEVs of cancer patients. It is highly encouraged that authors validate these findings in individual samples and demonstrate meaningful predictive performance. Moreover, how many of these methylated proteins were also identified in cancer cell line-derived sEVs?

Author Response

Point-by-Point Response 

We thank the reviewers for thoughtfully reviewing our manuscript. We appreciate their comments and suggestions, and have carefully addressed each in a point-by-point response.

Reviewer 1

In the current study, Bhandari and colleagues evaluated protein methylation patterns in circulating small extracellular vesicles from patients with early stage pancreatic cancer (N= 23) and matched controls. Authors found that symmetric dimethyl arginine (SDMA) in plasma sEVs was lower in patients with early or late stage PDAC; immunoprecipitation followed by proteomic analyses identified candidate arginine methylated proteins that were exclusively identified in sEVs from early-stage PDAC cases. Overall the study is interesting. However, enthusiasm is thwarted due to inadequate description of experimental conditions as well as lack of validation. 

 Patient plasmas. Authors state that “plasma from colon cancer patients was collected at the Stephenson Cancer Center”. Did the authors mean to say pancreatic cancer? While Tables 1 and 2 are fine, it would be easier to compare characteristics of cases and controls in a singular table that summarizes numbers as well as sex, age, and race distributions. Moreover, per text provide in results section, authors additionally included samples from patients with locally advanced colon cancer as well as patients with chronic pancreatitis. The source of these specimens needs to be detailed in the methods section.

Response: While we focused on plasma sEVs from patients with early stage pancreatic cancer, plasma sEVs from patients with late stage pancreatic cancer, colon cancer, and chronic pancreatitis were included as controls. The information about these patients was presented in Supple-Tables. We have emphasized the source of the plasma in the method section.      

Statistical analyses. Authors should detail which post hoc test was used for pairwise analyses following ANOVA.

Response: We have added the post hoc test in the figure legends to address this concern.

Figures 2-3. It is unclear if individual patient plasmas were used or if samples were pooled. Moreover, it is very difficult to ascertain how many samples were evaluated in the respective groups. This information is pivotal to make meaningful conclusions. Additionally, there are no error bars for the ‘healthy’ group, which suggests an N= 1. It is unclear how statistical significance would be derived in this setting. Individual data points show be provided for these representative bar plots. Figure C, band II appears to be oversaturated.

Response: We used individual plasma samples to generate the data presented in Figure 2-3. We modified the figure legends to make this clear. The reason we have no error bars for the health group is that for each Western blot band quantification, the intensity detected in individual healthy control was set as 100, and the intensity detected in individuals from other groups was related to the healthy individual. In this case, although we tested 16 samples in each group, there will be no error bars for the healthy group. We have modified the figure legend to make this clear.   

Figure 5. Proteins identified in SDMA band I and II. C3, CFH, A2M, and APOB are abundant plasma proteins and associated with inflammatory responses. This would suggest that these are not derived from cancer cells.

Response: We agree that these are abundant plasma proteins. However, overexpression of C3 and A2m in pancreatic cancer tissues was well described. In fact, these two proteins have been suggested as plasma biomarkers for pancreatic cancer (see references 30, 31, 46, 47) as we have discussed in the discussion section.

Validation of findings. Based on Figure 7C, there are several candidate proteins that are methylated and selectively identified in sEVs of cancer patients. It is highly encouraged that authors validate these findings in individual samples and demonstrate meaningful predictive performance. Moreover, how many of these methylated proteins were also identified in cancer cell line-derived sEVs?

Response: We appreciate the reviewer’s comments. We are actively recruit more plasma samples and ready to test each individual proteins listed in Figure 7 to determine their potential diagnostic value for early stage PDAC. We have not done proteomics with the sEVs derived from PDAC cell lines, partially because of the consideration that PDAC tissue-derived sEVs may be a better resource to compare with the patient plasma derived sEVs. We have collaborated with other research groups to obtain PDAC tissue-derived sEVs and will test the arginine methylation patterns of these sEVs when sufficient samples are collected.

Reviewer 2 Report

Comments and Suggestions for Authors

This study is interesting with clinical significance. Early stage pancreatic ductal adenocarcinoma is often insidious, and the symptoms are not obvious, but the progress is very fast. The opportunity for radical treatment is often lost. The authors put forward a new point of view on early diagnosis of early diagnosis of pancreatic cancer

The followings are comments to the authors:

1.Is exosome sEV? Why were exosomal markers((Flotillin-1, CD63 and CD9) ) used to identify sEV in Figure 1E?

2.Were all Patient plasma in the early stage group(n=16) or late stage groups(n=10) mixed together for Western blot analysis or each patient plasma tested separately?

3.There were 16 speciens in early stage group and late stage group in Table S2 in supplementary , but Sample size of clinicopathological features of early stage PDAC patients included in this study is 23 in table 1 (line 95). Please confirm that.

4. How to isolate sEV ?  Please state that in Materials and methods

5. The discussion and conclusion can be improved. These kinds of studies have limitations. Hence, the author should have stated the potential limitations and suggested what could be done the next step in this area of research.

6. Please state how many repeat trials/replicates of each method were conducted in the text and Figure legends?  Were the results consistent across all replicates?

7.The sample size in this study is small both in the early stage group(n=16) or late stage groups(n=10) , so I suggest expanding the sample size for verification in the further study. 

Author Response

Point-by-Point Response

We thank the reviewers for thoughtfully reviewing our manuscript. We appreciate their comments and suggestions, and have carefully addressed each in a point-by-point response. 

Reviewer 2

This study is interesting with clinical significance. Early stage pancreatic ductal adenocarcinoma is often insidious, and the symptoms are not obvious, but the progress is very fast. The opportunity for radical treatment is often lost. The authors put forward a new point of view on early diagnosis of early diagnosis of pancreatic cancer

The followings are comments to the authors:

1.Is exosome sEV? Why were exosomal markers((Flotillin-1, CD63 and CD9) ) used to identify sEV in Figure 1E?

Response: Yes, exosomes are the major components of small extracellular vesicles according to the ISEV (J Extracell Vesicles. 2018 Nov 23;7(1):1535750). The term sEV is often used nowadays because there is no exosome isolation methods that can guarantee to obtain pure exosomes. In the present study, we used the term sEV, and verified our isolates by exosome markers to show that exosomes are indeed major component of the sEVs.      

2.Were all Patient plasma in the early stage group(n=16) or late stage groups(n=10) mixed together for Western blot analysis or each patient plasma tested separately?

Response: Each patient plasma was tested individually for Western blot analysis.  We modified our figure legends to make this clear.

3.There were 16 speciens in early stage group and late stage group in Table S2 in supplementary , but Sample size of clinicopathological features of early stage PDAC patients included in this study is 23 in table 1 (line 95). Please confirm that.

Response: Yes, we have total 23 early stage PDAC plasma samples tested. In Table 1, we have noted which plasma samples were used for different experiments. 

4. How to isolate sEV ?  Please state that in Materials and methods

Response: We have described the sEVs isolation procedures in the materials and methods section.

5. The discussion and conclusion can be improved. These kinds of studies have limitations. Hence, the author should have stated the potential limitations and suggested what could be done the next step in this area of research.

Response: We agree. We have added a few sentences to discuss about the limitations and future directions of this study.

6. Please state how many repeat trials/replicates of each method were conducted in the text and Figure legends?  Were the results consistent across all replicates?

Response: We have listed the number for plasma samples used for each figure. Our Western blot was performed in three samples/group each round, and the results were generally consistent for each experiment.   

7. The sample size in this study is small both in the early stage group(n=16) or late stage groups(n=10) , so I suggest expanding the sample size for verification in the further study.

Response: We agree. We are actively recruiting plasma samples to expand our sample size and verify our findings.  

Reviewer 3 Report

Comments and Suggestions for Authors

The manuscript entitled Protein arginine methylation patterns in plasma small extracellular vesicles are altered in patients with early stage pancreatic ductal adenocarcinoma is an interesting and clinically significant observation presented by the authors. The manuscript is well conceptualized, and many experiments were carried out to support the findings. 

Authors have first validated the arginine methylation antibodies using the pancreatic cancer cell lines (total lysates and sEV lysates) and also validated the exosome from human plasma using a couple of markers. They further studied the arginine methylation pattern in healthy, colon cancer and early PDAC patients. Continuing the same, they also compared early and late PDAC patient samples and early PDAC and CP samples. Proteomic analyses of SDMA immunoprecipitated plasma sEV proteins from early-stage and healthy subjects were performed to prove arginine methylation patterns as a differentiator/marker of early-stage PDAC. 

Certain points need to be addressed before the manuscript can be considered for publication: 

The manuscript needs thorough re-checking and improvement in writingThere are many grammatical mistakes and some sentences do not explicitly convey the right message. A few examples are mentioned below:  

  1. Sensitive and specific circulating biomarkers for early detection of pancreatic ductal adenocarcinoma (PDAC) are urgently needed to improving survival outcomes of this malignancy. Correct the sentence Sensitive and specific circulating biomarkers for early detection of pancreatic ductal adenocarcinoma (PDAC) are urgently needed to improve the survival outcomes of this malignancy. 

  1. The sentence Studies have shown that plasma sEV molecules, such as proteins and microRNAs, are potential indicators of PDAC is misleading. Reframe it. 

  1. Correct the sentence in the abstract “However, previous efforts have been largely focused on proteomics and miRNA signatures in plasma sEVs. The post-translational modifications of sEV proteins, such as arginine methylation, have not been explored. 

  1. “In this context, posttranslational modifications of plasma sEV proteins, such as arginine methylation patterns, have never been studied as circulating biomarkers for PDAC.” Reframe the sentence as “In this context, posttranslational modifications of plasma sEV proteins, such as arginine methylation patterns, have been explored as a potential circulating biomarker for PDAC.” 

  1. How do the authors explain the use of only the samples from 2 specific races, CA and AA patients? Why not the larger cohort including other races? 

  1. Explain the purpose and significance of comparing the colon cancer patient samples with the early PDAC samples. 

  1. SDMA bands are significantly reduced in both the early-stage PDAC and late-stage PDAC. How will a clinician differentiate between the stages using these as biomarkers? 

  1. The authors should also include the samples from late-stage PDAC patients for the proteomic analyses to bring out the difference between the arginine methylated proteins in both stages.

Comments on the Quality of English Language

English language needs extensive editing.

Author Response

Point-by-Point Response

 We thank the reviewers for thoughtfully reviewing our manuscript. We appreciate their comments and suggestions, and have carefully addressed each in a point-by-point response.

Reviewer 3

The manuscript entitled “Protein arginine methylation patterns in plasma small extracellular vesicles are altered in patients with early stage pancreatic ductal adenocarcinoma” is an interesting and clinically significant observation presented by the authors. The manuscript is well conceptualized, and many experiments were carried out to support the findings. Authors have first validated the arginine methylation antibodies using the pancreatic cancer cell lines (total lysates and sEV lysates) and also validated the exosome from human plasma using a couple of markers. They further studied the arginine methylation pattern in healthy, colon cancer and early PDAC patients. Continuing the same, they also compared early and late PDAC patient samples and early PDAC and CP samples. Proteomic analyses of SDMA immunoprecipitated plasma sEV proteins from early-stage and healthy subjects were performed to prove arginine methylation patterns as a differentiator/marker of early-stage PDAC. Certain points need to be addressed before the manuscript can be considered for publication: 

The manuscript needs thorough re-checking and improvement in writing. There are many grammatical mistakes and some sentences do not explicitly convey the right message. A few examples are mentioned below:  

  1. “Sensitive and specific circulating biomarkers for early detection of pancreatic ductal adenocarcinoma (PDAC) are urgently needed to improving survival outcomes of this malignancy.” Correct the sentence “Sensitive and specific circulating biomarkers for early detection of pancreatic ductal adenocarcinoma (PDAC) are urgently needed to improve the survival outcomes of this malignancy.”

Response: The manuscript has been carefully reviewed by an experienced English-speaking scientist, and we have modified the sentence accordingly.

2. The sentence “Studies have shown that plasma sEV molecules, such as proteins and microRNAs, are potential indicators of PDAC” is misleading. Reframe it.

Response: We have modified this sentence.

4. Correct the sentence in the abstract “However, previous efforts have been largely focused on protein and miRNA signatures in plasma sEVs. The post-translational modifications of sEV proteins, such as arginine methylation, have not been explored.”

Response: We have modified this sentence.

4. “In this context, posttranslational modifications of plasma sEV proteins, such as arginine methylation patterns, have never been studied as circulating biomarkers for PDAC.” Reframe the sentence as “In this context, posttranslational modifications of plasma sEV proteins, such as arginine methylation patterns, have been explored as a potential circulating biomarker for PDAC.”

Response: We have modified this sentence.

5. How do the authors explain the use of only the samples from 2 specific races, CA and AA patients? Why not the larger cohort including other races?

Response: This is determined primarily by the sample availability. Early stage PDAC plasma samples are difficult to acquire. We are actively recruiting more plasma samples for this project in collaboration with the NCI sponsored CHTN. 

6. Explain the purpose and significance of comparing the colon cancer patient samples with the early PDAC samples.

Response: The colon cancer plasma samples are included to serve as another group of control, which may provide indications as to whether the arginine methylation patterns are PDAC specific. 

7. SDMA bands are significantly reduced in both the early-stage PDAC and late-stage PDAC. How will a clinician differentiate between the stages using these as biomarkers?

Response: This is a study in its discovery stage. The difficulty of detecting early stage PDAC is largely due to its asymptomatic nature and lack of clinical biomarkers.  Late stage PDAC is often manifested with clinical symptoms, detected by image examinations, and confirmed by biopsy. Should the arginine methylation patterns be verified as indicators of early stage PDAC in large cohort studies, it will help screen populations with high risk for PDAC (Those with family PDAC history, diabetes, obesity, heavy drinkers, and pancreatic cysts) to detect PDAC earlier.

8. The authors should also include the samples from late-stage PDAC patients for the proteomic analyses to bring out the difference between the arginine methylated proteins in both stages.

Response: Thanks for the suggestion. We are planning to profile arginine methylated proteins in plasma sEVs derived from patients with late stage PDAC, which may provide key information enable us to differentiate early stage from late stage PDAC. 

Round 2

Reviewer 1 Report

Comments and Suggestions for Authors

Title: Protein arginine methylation patterns in plasma small extracellular vesicles are altered in patients with early stage pancreatic ductal adenocarcinoma

This Reviewer appreciates changes made by the authors to address prior concerns. However, there are still a few issues that need to be considered.

11)   Post-hoc analyses. Authors should clearly stage what type of post-hoc analysis was used for multiple comparison testing (i.e. Tukey, Dunn’s, other). This information should also be included in the methods and not limited to the figure legends. 

22) Western blots. Authors state that for healthy controls, the bar was set as 100%. However, in doing so, this fails to capture the variation that may occur in otherwise healthy individuals. It would be much better to show immunoblots for several healthy controls as well as cases to capture heterogeneity.

33) Tissue EVs vs PDAC cell line derived EVs. This Reviewer acknowledges the authors comment regarding PDAC tissue derived sEVs compared to cell line derived sEVs as valid. Nonetheless, tumors are a heterogenous mixture of cell types, each of which can contribute to the overall sEV profiles. This is particularly relevant given the lack of specificity of markers such as C3. CFH, A2M, and APOB. Use of cancer cell-derived sEVs will provide additional confidence that signature features are more related to pancreatic cancer cells rather than (for instance) tumor infiltrating immune cells.

Author Response

We appreciate this reviewer’s positive comments and thoughtful suggestions.

11) We have included the information about our post-hoc test for multiple comparisons in the Materials and Methods section.

22) We thank the reviewer for this comment, and will carefully consider the reviewer’s suggestion in our future quantification of the Western blot data.

33) This is a very thoughtful comment. We should include both cancer tissue-derived and cancer cell-derived sEVs for our future analysis.

Reviewer 2 Report

Comments and Suggestions for Authors

This manuscript can be accepted in present form

Author Response

We appreciate this reviewer’s time and effort in reviewing our manuscript.

Reviewer 3 Report

Comments and Suggestions for Authors

The authors have addressed the comments to a major extent. I would recommend the acceptance of the manuscript in its current form.

Author Response

(The authors gave the same response as above.)
